# Wafer-Scale Epitaxial Low Density InAs/GaAs Quantum Dot for Single Photon Emitter in Three-Inch Substrate

**DOI:** 10.3390/nano11040930

**Published:** 2021-04-06

**Authors:** Xiaoying Huang, Rongbin Su, Jiawei Yang, Mujie Rao, Jin Liu, Ying Yu, Siyuan Yu

**Affiliations:** 1State Key Laboratory of Optoelectronic Materials and Technologies, School of Electronics and Information Technology, Sun Yat-sen University, Guangzhou 510275, China; huangxy327@mail.sysu.edu.cn (X.H.); yangjw8@mail2.sysu.edu.cn (J.Y.); raomj@mail2.sysu.edu.cn (M.R.); 2State Key Laboratory of Optoelectronic Materials and Technologies, School of Physics, Sun Yat-sen University, Guangzhou 510275, China; surongbin@mail.sysu.edu.cn (R.S.); liujin23@mail.sysu.edu.cn (J.L.); 3Photonics Group, Merchant Venturers School of Engineering, University of Bristol, Bristol BS8 1UB, UK

**Keywords:** three-inch wafer-scale, low density QD, CBR-HBR

## Abstract

In this work, we successfully achieved wafer-scale low density InAs/GaAs quantum dots (QDs) for single photon emitter on three-inch wafer by precisely controlling the growth parameters. The highly uniform InAs/GaAs QDs show low density of $0.96/{\mu m}^{2}$ within the radius of 2 cm. When embedding into a circular Bragg grating cavity on highly efficient broadband reflector (CBR-HBR), the single QDs show excellent optoelectronic properties with the linewidth of 3± 0.08 GHz, the second-order correlation factor $g^{2}\left(\tau \right)=0.0322\;\pm 0.0023$, and an exciton life time of 323 ps under two-photon resonant excitation.

## 1. Introduction

Semiconductor quantum dots (QDs), due to its discrete energy levels as artificial atoms, serve as a core element in the emerging application of optoelectronic devices including lasers [1], solar cells [2], and photodetectors [3]. The rapid development of quantum computing [4], quantum cryptography [5], as well as quantum key distribution (QKD) [6] in recent years, busting many researches in low-density quantum dots for the generation of ideal single-photons and entangled-photon pairs via external optical/electrical pulses [7,8]. Last three decades have witnessed the rapid development of QDs from concept to reality via advanced molecular beam epitaxial technique, including Stranski–Krastanow (S–K) mode growth [9], droplet epitaxy [8,10], as well as site-controlled growth [11,12]. Regarding QD production methods, scalability is very important that allowed production of individual, identical QDs deterministically at specific locations on a substrate, and emitting highly coherent, identical photons at exactly the same energy. Currently, however, the QD production methods are often in randomly positioned and highly inhomogeneous QD populations, which strongly impairs the deterministic production of devices based on single QDs, posing a steep challenge to scalability. Site-controlled growth, which addresses spatial randomness, has suffered from defects in previously processed surfaces which diminish the quantum efficiency and coherence of emitted photons [13]. For the S–K growth, In(Ga)As/GaAs QD-based devices have shown great performance as quantum emitters with close to unity quantum efficiency [14,15] and near transform-limited emission [16]. However, reliable wafer-scale growth techniques have proved elusive. Low density less than 1 $\mathrm{dot}/{\mu m}^{2}$ is a crucial element for single photon emitter, which needs to precisely control of the experimental parameters including growth temperature [17], deposition rate [18], deposition amounts [19], As flux [20], and III/V ratio [21]. A gradient InAs quantum dot growth technique, which via stopping substrate rotation and applying an Indium flux gradient to form a gradient density of InAs QDs [19,22,23], has been considered the most effective way to achieve low density InAs QDs. However, with the drawback of low-yield of this technique, it’s of great importance to improve the utilize region for single quantum dots of wafer scale via precisely controlling experimental condition.

In this work, we successfully obtained wafer-scale InAs/GaAs QDs for single photon emitter with the density as low as $0.96/{\mu m}^{2}$ on a three-inch wafer by precisely controlling the growth temperature, InAs deposition amount and capping GaAs thickness through monitoring the temperature distribution of the substrate. We noted that due to the limitation of our system that the substrate heater has only one heating filament, there is a temperature distribution over the three-inch wafer which is the one responsible for nonuniformity of QDs. However, it provides a feasible approach to achieve wafer-scale uniform low density single QDs for other growth systems that can uniformly control the thermal distribution of the whole wafer. The single QDs show excellent optoelectronic properties with the linewidth of 3± 0.08 GHz, the second-order correlation factor $g^{2}\left(\tau \right)=0.0322\;\pm 0.0023$, and an exciton life time of 323 ps under two-photon resonant excitation, when coupling to a circular Bragg grating cavity.

## 2. Method

Our samples were grown on semi-insulating three-inch GaAs (001) substrates in a solid-source MBE chamber equipped with high-energy electron diffraction (RHEED) and a cracker cell for as evaporation. The details of experimental parameters including the growth temperature of InAs layer, the thickness of GaAs capping layer, the deposition rate and amount of InAs, and the As flux beam equivalent pressure, are listed in Table 1. The deposition temperatures are calibrated by the transition temperature *Tc* (as measured by thermocouple) when the reconstruction pattern of GaAs surface seen in RHEED transfers from (2 × 4) to (2 × 3) [24]. In details, the substrates were degassed at 680 °C for 10 min under an overpressure of as prior to growth. A 300 nm GaAs buffer layer was grown at 660 °C at a growth rate of 0.6 μm/h with an As flux of 4.8 × 10^−6^ Torr beam equivalent pressure. This was followed by the first InAs QD layer growth, which was stopped at a critical coverage for island formation (*θc*) monitored via RHEED pattern [25,26] (seen in Appendix A). Then the substrates were annealed at 680 °C for 10 min to evaporate the indium atoms completely and subsequently grown with 90 nm GaAs and the second QD-layers. In details, as shown in Figure 1, Sample A was designed to fabricate single layer of high density InAs QDs for the measurement of temperature distribution of the three-inch wafer. It was deposited with 150%$\theta_{C}$ using high Indium growth rate of 0.08 ML/s at the temperature of (Tc-40) °C. Samples B–I were designed to optimize parameters to achieve three-inch wafer-scale low density InAs QDs, they were deposited for Photoluminescence (PL) measurement by using ultra-low Indium growth rate of 0.004 ML/s at the temperature of (Tc-25) °C. The high substrate temperature and low deposition rate increase migration length of In atoms to achieve low density quantum dots [27]. The variation in deposition amount of 100%$\theta_{C}$, 90%$\theta_{C}$, 85%$\theta_{C}$, 83%$\theta_{C}$, 80%$\theta_{C}$ in samples B–F results in different QD density. While the variation in the thickness of GaAs capping layer in samples G–I results in different QD emission wavelength. Then the flushing technique [28] of InAs QDs is introduced to tune the size of QDs, which was ongoing at 680 °C for 3 min under arsenic flux. An additional InAs QDs layer without indium flushing was deposited on the top of 90 nm GaAs spacing layer for the morphology measurement by atomic force microscopy [AFM, Bruker (Santa Barbara, CA, USA), Dimension Icon in peak force tapping mode], using the same parameters of substrate temperature, asenic flux, InAs deposition amount. The QD densities in AFM images (1 μm ×1 μm) are calculated via the software of AFM Nanoscope Analysis, then the substrates were cooled to room temperature immediately with the rate of 100 °C/min.

## 3. Results and Discussion

Substrate temperature is one of the most pivotal parameters in the growth of low density InAs QDs as it effects the adsorption, desorption, migration processes of In atoms [27]. To achieve wafer-scale samples, we firstly investigate the temperature distribution of the three-inch substrate in our growth system. High density InAs QDs grown at lower temperature without Indium desorption [29] (Sample A) are used to comprehensively calibrate the temperature in the whole wafer. Figure 2b,c are the experimental results from sample A, Figure 2b illustrates representative AFM results of InAs QDs along [110] crystallographic direction from left to the right of the wafer with 5 mm interval distance. As shown in (4)–(11), the QD density increases gradually from center to the edge within the radius of 2 cm, but increase more rapidly from 2 cm to the edge. Other crystallographic directions including [1–10], diagonal 1, diagonal 2 demonstrate the same regularity as well, as shown in Figure 2c. These results indicate that there is a temperature distribution over the three-inch wafer. Hence eight other high density QD samples grown at different temperatures of 567 °C, 562 °C, 557 °C, 552 °C, 547 °C, 542 °C, 537 °C, and 532 °C are fabricated to determine this distribution by investigating their QD densities in the same region (detailed AFM results are in Appendix A). Regardless of indium desorption at lower temperature, we found that the QD density increases almost linearly as a function of the deposition temperature in Figure 2d. Thus, the temperature distribution of the whole three-inch wafer is deduced, as shown in Figure 2e. In the radius of 2 cm as depicted in the inset image, the temperature is stable and about 8 °C lower than that of the center. While it decreases more rapidly and about 25 °C lower from the position of +3.5 cm to the center, and about 30 °C from the position of −3.5 cm to the center. The left side edge decreases more rapidly than the right one is mainly due to the more defect at the cut edge leading high thermal conductivity. 

After confirming temperature distribution, the critical coverage *θc* for 2D to 3D transition is precisely determined from the RHEED signals (as seen in Appendix A). As shown in Table 2, a nearly linear increase of *θc* with temperature range from (Tc-40) °C to (Tc-20) °C contradicts a simple thermally activated process. When the temperature increases to (Tc-15) °C, we found there is no 3D transition, that is in reasonable agreement with temperature-dependent indium segregation and desorption during growth. To increase the repeatability of the growth process, the first InAs layer was grown at a temperature of (Tc-25) °C for in situ determining the *θc*. Different QD densities are investigated by changing the InAs deposition amounts from 100%$\theta_{C}$ to 80%$\theta_{C}$ in Samples B–F. Typical AFM images at the central of the wafer are shown in Figure 3a. Only small dots below 10 nm height observed at the deposition of 100%$\theta_{C}$, 90%$\theta_{C}$ and 85%$\theta_{C}$ with domain QD height in 6 nm, 4 nm, 5 nm, respectively. Average height and density of small QDs decrease as the deposition amount decrease. The insert graphic shows some larger dots above with height in 7–10 nm. When comes to the deposition of 83%$\theta_{C}$ and 80%$\theta_{C}$ only small dot below 6 nm and 5 nm demonstrated, separately. We observed another phenomenon with bimodal dot distribution (with larger dots above 12 nm height and small dots below 8 nm height) in 83%$\theta_{C}$, which we consider that it is mainly caused by different diffusion length of Indium atoms in different substrates (Seen in Appendix A). Figure 3c illustrates the micro-PL spectra of Samples B–F in the central of the wafer. In the deposition of 80%$\theta_{C}$, none isolated lines observed, indicating the ultra-low confinement of electrons and holes of QDs below the height of 5 nm. A single line emits from single dot at wavelength centered around 900 nm observed in 83%$\theta_{C}$, we note that the QD emission is rare in the center part of this sample as it takes a long time to find a emission line, thus we believe QD height of 6 nm do no contribution to the emission. The broad peaks in Samples B–D are identified as the emission from the 7–8 nm height InGaAs QDs correspond to previous study [20]. We suppose that large QDs with height above 8.5 nm have been decomposed during indium flushing, which approved by the reconstruction pattern seen in RHEED transfers to (2 × 4) again. Additionally, we observed a flat surface with roughness of 0.2 nm measured by AFM with an extra sample which stopped growth after the process of indium flushing (as seen in Appendix A). 

We further optimize the uniformity of QD by investigating the influence of the thickness of GaAs capping layer during Indium flushing process [30,31]. The thickness of GaAs capping layer in Sample G-I are 6.5 nm, 4.5 nm, 2.5 nm, respectively. We note that there are no emission lines in Sample I with 2.5 nm GaAs capping layer as found in Sample F with 80%$\theta_{C}$ deposition amount. Figure 4 summarizes PL measurements of ~120 QDs of Sample E, G and H, demonstrating the uniformity of the emission wavelength. A PL broadening of 11 meV was found for Sample H. This is slightly smaller than that achieved by the S–K QD growth in Sample E. The blueshift of the wavelength with thinner GaAs cap layers is probably caused by the decreased dot size via the flushing technique, which influences the band gap of the single QDs [31,32]. 

Now we turn to study the density distribution of sample H in the wafer-scale. We divide the three-inch wafer into seven regions from −3 cm to +3 cm with 1 cm interval distance along [110] crystallographic direction. Figure 5a shows representative AFM images and the corresponding statistical QD height distribution in (b). Due to short migration length of In atoms, the density of 7–8 nm QDs emitted around 900 nm increases from center to edge of the wafer as the temperature decreases [33]. Typical emission spectra of the QDs under above-barrier excitation using a continuous wave 785 nm laser, corresponding with the wide-field silicon electron multiplied charged couple device (Si-EMCCD) images of the fluorescence from the response samples, are described in Figure 5d,e. Figure 5d illustrates fluorescent images from QDs measured under red LED (730 nm) illumination, we count the QD density of the blight dots in the image. Within the radius of 2 cm we got a large scale of individual QDs with sharp emission lines, from where −2 cm to +2 cm with QD density of $0.96/{\mu m}^{2}$, $8.5\times 10^{-2}/{\mu m}^{2}$, $1.3\times 10^{-3}/{\mu m}^{2}$,$\;9.6\times 10^{-3}/{\mu m}^{2}$
$5.8\times 10^{-2}/{\mu m}^{2}$, respectively. This would be a popular QD density region for further nanostructure fabrication and optical positioning [34]. Figure 5e shows the representative ensemble PL spectra when the 785 nm laser (with faculae radius about several microns) excites the QDs on the seven regions. The number of emission lines increase from the center to the edge as the density in AFM illustrated. We should note that small dots below 6 nm would not be illuminated [20].

For single-photon purity assessment, we encapsulate our single QD of Sample H (in the region 2 in Figure 5) into the state-of-art nanostructure circular Bragg grating cavity on highly efficient broadband reflector (CBR-HBR) as the same process of previous study [35]. Figure 6a illustrates the structure of CBR-HBR which consists of a circular GaAs disk surrounded by a set of concentric GaAs rings, sitting on a SiO_2_ layer with a back-reflector consisting of a gold layer as previously study [36]. Photoluminescence spectrum of QD emit at ~905 nm under two-photon resonant excitation and emission narrow line width fitted by Voigt curves of 3± 0.08 GHz in Figure 6b. Second-order autocorrelation measurement, under ‘π pulse’ two-photon resonant excitation by using a Hanbury–Brown and Twiss set-up, shows a low value of $g^{2}\left(\tau \right)=0.0322\;\pm 0.0023$ in Figure 6c. The nearly absence of coincidence events at zero time delay indicates the high purity of the emitted single photons. The photoluminescence decay data in Figure 6d fitted by monoexponentially decay function, allowing us to extract a lifetime of 323 ps.

## 4. Conclusions

In summary, through carefully investigating the thermal distribution of three-inch wafer, as well as precisely controlling the InAs deposition temperature, deposition amount and GaAs capping layer, we have achieved uniform emission wavelength of low density InAs/GaAs QDs that suitable for nanofabrication within the radius of 2 cm. By embedding the QD into CBG-HBR, a pure and bright single quantum emitter with the second-order correlation $g^{2}\left(\tau \right)=0.0322\;\pm 0.0023$ and lifetime of 323 ps are demonstrated under two-photon resonant excitation. We noted that if the growth system can uniformly control the temperature distribution of the whole wafer, wafer-scale uniform low density single QDs on three-inch wafer and beyond can be achieved. It will pave a way for the generation of scalable quantum light sources [37]. 

## Figures and Tables

**Figure 1 nanomaterials-11-00930-f001:**
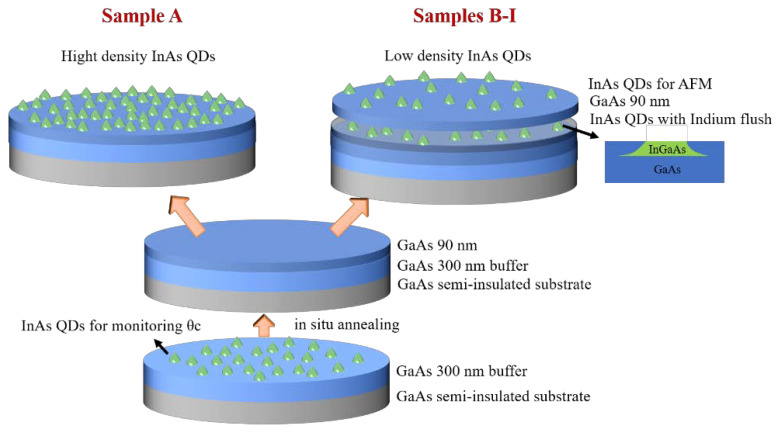
Schematic process of the quantum dot (QD) growth: Sample A was designed to fabricate single layer of high density InAs QDs for the measurement of temperature distribution of the three-inch wafer, while Samples B–I were designed to optimize parameters to achieve three-inch wafer-scale low density InAs QDs.

**Figure 2 nanomaterials-11-00930-f002:**
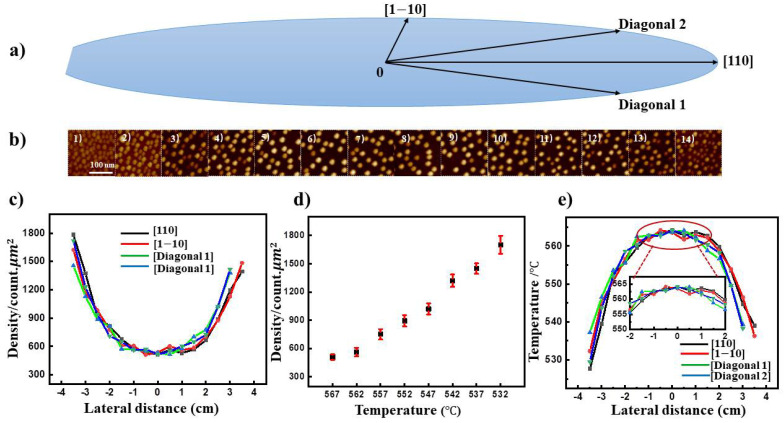
(**a**) three-inch wafer geometric structure: crystallographic directions of [110] and [1–10], diagonal 1 and diagonal 2; 0 represents the geometric center of the wafer, the arrow direction represents the positive direction, respectively. (**b**) Representative AFM results of InAs quantum dots of sample A in [110] crystallographic direction: (1)–(14) from left to right with 5 mm interval distance. (**c**) Density distribution of the four directions of sample A of the whole wafer, black line represents [110], red line represents [1–10], green line represents diagonal 1, blue line represents diagonal 2, respectively. (**d**) The density of quantum dots as a function of substrate temperature with red error bars. (**e**) wafer-scale temperature distribution of three-inch wafer with inset graphics shows small fluctuation of temperature from −2 to 2 cm.

**Figure 3 nanomaterials-11-00930-f003:**
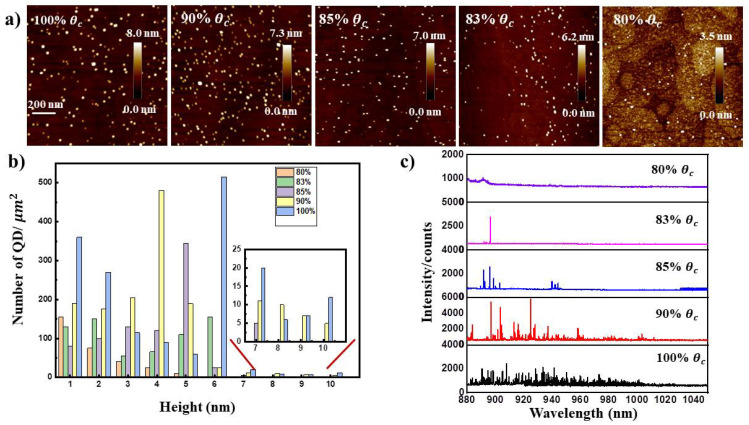
(**a**) Representative of AFM images of 1 μm ×1 μm in center part of three-inch wafer with different InAs deposition of 100%$\theta_{c}$, 90%$\theta_{c}$, 85%$\theta_{c}$, 83%$\theta_{c}$, 80%$\theta_{c}$, respectively; (**b**) Statistical height distribution of QDs in the region of 5 μm ×5 μm, corresponding ensemble photoluminescence spectra from sample B to F with emission wavelength rage from 880 nm to 1040 nm in (**c**).

**Figure 4 nanomaterials-11-00930-f004:**
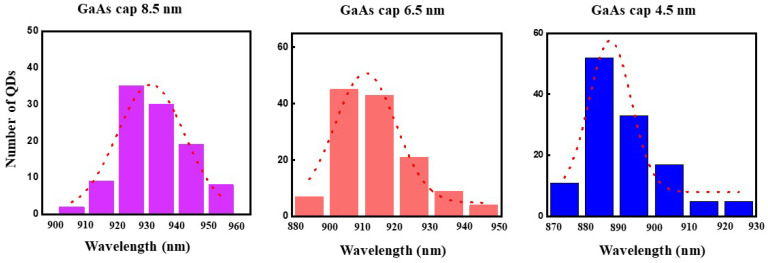
Statistical wavelength distribution of about 120 single QDs with different GaAs capping layer of 8.5 nm, 6.5 nm and 4.5 nm in sample E, G, H, respectively.

**Figure 5 nanomaterials-11-00930-f005:**
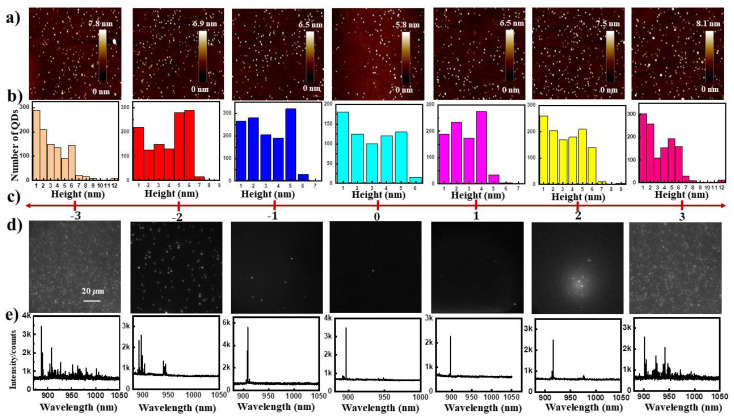
Both topographic, optical and statistical QD height distribution of sample H along [110] crystallographic direction, which is divided into seven regions from −3 to +3 with 1 cm interval distance, (**a**) Representative AFM images in 1 μm ×1 μm, (**b**) Statistical height distribution of QDs in 5 μm ×5 μm, (**c**) The seven regions, (**d**) Example fluorescent images from QDs measured under red LED (730 nm) illumination, a 800-nm long-pass filter (LPF) is inserted into the collection path when measuring the QD images shown in EMCCD, (**e**) the representative ensemble PL spectra under 785 nm laser illumination.

**Figure 6 nanomaterials-11-00930-f006:**
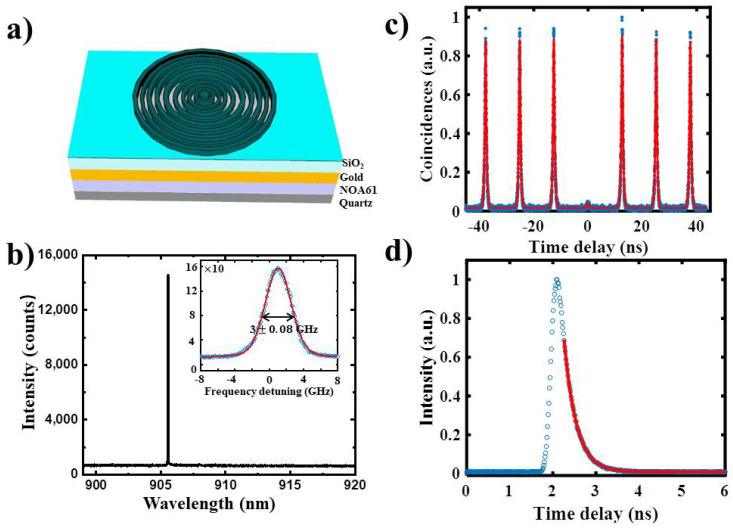
Optical properties of QD in circular Bragg grating cavity on highly efficient broadband reflector (CBG-HBR). (**a**) Schematic of CBR-HBR. (**b**) Photoluminescence spectra of a QD in CHG-HBR by two-photon resonant excitation, the insert image show the emission line width. (**c**) The second-order correlation $g^{2}\left(\tau \right)=0.0322\;\pm 0.0023$ measured under pulsed 857 nm excitation, using a Hanbury–Brown and Twiss setup. (**d**) Decay time measurements, the excited state decays (blue dots) fitted by monoexponentially curves (red-solid lines).

**Table 1 nanomaterials-11-00930-t001:** List of experimental parameters of investigated samples.

Sample ID	Tc Thermocouple (°C)	T (InAs)/ Thermocouple (°C)	Thickness of GaAs Cap Layer (nm)	In Deposited Rate (ML/s)	In Deposited Amount $\left(\theta_{C}\right)$	As Flux (Torr)
A	602	562	0	0.08	150%	1.8 × 10^−6^
B	605	580	8.5	0.004	100%	5 × 10^−7^
C	603	578	8.5	0.004	90%	5 × 10^−7^
D	603	578	8.5	0.004	85%	5 × 10^−7^
E	605	580	8.5	0.004	83%	5 × 10^−7^
F	602	577	8.5	0.004	80%	5 × 10^−7^
G	600	575	6.5	0.004	83%	5 × 10^−7^
H	605	580	4.5	0.004	83%	5 × 10^−7^
I	605	580	2.5	0.004	83%	5 × 10^−7^

**Table 2 nanomaterials-11-00930-t002:** Optimization of $\theta_{C}$ under various substrate temperatures.

T (Calibration)/°C	T (InAs)/Thermocouple (°C)	In Deposited Rate (ML/s)	As Flux (Torr)	InAs Deposited Amount/ML
(Tc-40) °C	562	0.004	5 × 10^−7^	1.60
(Tc-30) °C	572	0.004	5 × 10^−7^	1.67
(Tc-25) °C	577	0.004	5 × 10^−7^	1.86
(Tc-20) °C	582	0.004	5 × 10^−7^	1.92
(Tc-15) °C	587	0.004	5 × 10^−7^	Not appear

## Data Availability

The data of this study are available within this article, further inquiries may be directed to the authors.

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
