# Peer review of "Wafer-Scale Epitaxial Low Density InAs/GaAs Quantum Dot for Single Photon Emitter in Three-Inch Substrate"

_nanomaterials, 2021, doi:10.3390/nano11040930_

Round 1
Reviewer 1 Report
The authors demonstrated wafer scale low density InAs/GaAs quantum dots (QDs) for single photon emitter on 3-inch wafer. This is an important achievement in the field that can be used by everybody to move forward toward cheaper single photon sources. I think the paper can be publihsed as it is.
Author Response
We thank the reviewer's very much for highly aggreement of our study, we've sumited a new version with more detail and logical configuration, we hope our current version is more suitable for publication in Nanomaterials.
Reviewer 2 Report
In this manuscript, a systematic study on the distribution of epitaxial InAs/GaAs quantum dots is conducted on the wafer-size, in order to achieve uniform, low-density QDs for single-photon emitter applications. While the overall idea is clear, the description of the experimental growth strategy and characterization, which is the key of the present study, is critically insufficient so that many aspects are obscure to the reader and make it hard to understand key points. For this reason, the manuscript is not suitable for publication and requires in-depth major revisions to address the following points:
1) The Methods section does not provide all necessary information on the growth process. Details are provided for the GaAs buffer growth and the subsequent InAs deposition. Then, things get confused as in line 81 it is said that samples B-I (but not A?) follow the scheme in figure 2a (why not figure 1?), which is really not explained in the caption and seems to indicate multiple layers of InAs QDs (red triangles?), including an in-situ annealing after the first InAs deposition of which there is no mention in the text. Line 81 also refers to the table S1 in supporting material which however is not commented and only adds a column for the temperature calibration. Finally, the last lines of the Method section state that AFM measurements are performed using an additional InAs deposition of which no detail is reported. The authors should write in detail all processes and growth stages they perform, clarify at which stage they perform the measurements and set a consistent schematic drawing. As the main achievement of the paper is the identification of the optimal conditions to achieve the targeted uniformity and low density, a precise indication of the growth apparatus and of the exploited parameters is mandatory to ensure reproducibility of the reported experiments.
2) From the previous point, in the last lines of the Methods section it is said that AFM measurements are performed using an additional InAs deposition of which no detail is indicated. Is this "second" layer of QDs the one imaged in all AFM images in the paper? Are these QDs the ones of interest for the present analysis or (as it seems by reading the methods) the key ones are the QDs formed after the first InAs deposition? In the second case, can the author explain how the QDs on the top layer can give reliable information on the underlying ones?
3) The measurement of the QD density is at the core of the whole study but it is not indicated how it is determined. Reasonably it comes form a counting in AFM images but it should be clarified along with some information on how representative and accurate is the estimate (sampled area, number of QDs, repetions of the measurements on different areas and different samples)
4) In the panel (d) of figure 1 it is shown how the density changes with temperature but figure 1 refers to the sample A, which is grown at a specific temperature. Looking to the figure S2 in the supporting information it appears (from the only information provided in the caption) that multiple samples were grown at different temperatures, which is never mentioned in the main text. Please clarify.
5) In lines 135 and 170, a bimodal distribution of the dot sizes is indicated but it is hardly noticeable, at least from the AFM views of Fig. 2b or Fig. 3a where dots are not clearly distinguishable by eye. Also the additional Table S2 in the supporting information doesn't really give a proof of this bimodal distribution but only indicates the number of QDs above and below an height of 8nm. The authors should plot the actual size (height) distribution for the dots in the sampled area to give a more convincing evidence that it is bimodal. Similarly, it is hard to say that by the AFM views of Figure 2b the reader can understand much on height and density as stated in line 134. A better elaboration of the data, e.g. by building the height distribution could be more clear.
6) Can the author clarify the sentence in line 140-142?
7) In lines 145-147 it is commented that the surface of sample E becomes flat possibly because of island decomposition after In flushing. An AFM image of sample E should be present in Fig. 2b (with QDs), which is totally different from the AFMs in the indicated Fig. S3 of the supporting material. Are these AFM taken at different stages (e.g. before and after flushing?). Incidentally, what is the image (9) of figure S3?
8) The authors should add a bit more details about the fluorescence measurement of Figure 3c-d and state more clearly the difference in QD density with respect to the AFM one.
9) Supporting information in the present form is not very helpful to the reader as few words are said only in the figure caption and most of the meaning is left to the intuition of the reader. If non trivial information are included they should be complemented by a minimum of explanation
10) The general english level is poor. Many grammar mistakes are present and some sentences (e.g. the already cited lines 140-142) are unclear in the meaning. A careful language revision is required.
Author Response
Please see the attachement.

Reviewer 3 Report
The paper focuses on defining a parameter range ensuring growth on relatively large areas of low-density InAs/GaAs QDs. While there is no fundamental novelty demonstrated, having the suitably low density and emission wavelength within a larger area is a technology relevant result with possible impact on applications. While there are many papers reporting optimization of QD growth towards low density, the typical growth fulfilling such low densities results in small areas (typically demonstrated on 2inch wafers) . The parameters range tested may provide useful hints for further optimizations. The optical results show that the QDs are good quality regarding the quantum optics applications, so the assumptions made in motivation sections are verified. The technical descriptions and methodology appear adequate.
Regarding the language style, some of the sentences are rather long (in particular in the introduction part). Then there are some typos, e.g. use of “molecule beam” instead of “molecular beam” that should be corrected.
Author Response
We thank you very much for your highly aggreement of our study, regarding to the language style, we've carefully revised in the new version and added more details about the experiemetal section , we hope our current version is suitable for publication in Nanomaterials.

Round 2
Reviewer 2 Report
In the revised version of the manuscript, the authors clarified some of the obscure points related to the growth process but there are still key aspects which are not clear. Then, major revisions are still needed. Here below the key points:
1) the height statistics taken from AFM analysis of samples at 83%Theta_c does not prove at all that there is a bimodal distribution but only shows that small QDs (below 8.5nm) are present. Changing the binning in the histogram before and after 8.5nm is not correct as it makes impossible to distinguish if there is a real secondary peak for a certain small QD height or if the distribution is just broadened toward small sizes (e.g. because of late nucleation events or coarsening). Secondly, the peak below 8nm accounts for less than 10 QDs which is not statistically sufficient. If having a bimodal distribution is key the authors should find a way to distinguish the QD sizes with the same binning as above 8.5nm and accumulate more statistics e.g. by analysing larger areas or multiple samples. At present, all references to bimodal distribution should be removed.
2) When considering samples B-F, the only difference should be the amount of InAs deposition. Is it then correct to think that the AFM images of the topmost QDs in Figure 3a correspond to a sort of time series i.e. the sample at 100%theta_c is the evolution of the one at 80% -> 83% -> 90% if deposition were not interrupted. Then, it is a bit of surprise to find that for 83% there exist QDs which are much larger than those observed at later stages (90% and 100%). Can the authors give an explanation to this? The manuscript says that the observed distribution can be "attributed to high migration length and inevitably in situ gathering of In atoms at the condition of high substrate temperature" but this should hold true also for greater coverages if temperature is the same. A missing information (again to be clarified in the methods) is how fast this QDs are cooled down so to freeze their morphology as the one of those being flushed in the layer underneath.
3) it is not yet clear which are the "active" QDs in the PL measurements. The ones buried after flushing or those on the topmost layer imaged by AFM? According to the Methods section the last uncapped QDs should be only for the AFM analysis so, is the PL performed before their deposition? Please write this more clearly to avoid misunderstandings.
4) Closely related to 3) and not answered from the previous comments. In Figure 4, the authors combine AFM images and PL fluorescence map showing rather different density of QDs. I suppose that the PL density refers to the active QDs but, as it is much smaller than the one of AFM images, it should be explained. Is it the total density of QDs in the buried layer or it accounts only for those emitting in the selected wavelength region?
5) The explanation of the single-photon measurements (Fig. 5) is minimal. Can the authors clarify how they performed the measurement in practice, i.e. how they locate and embed the single QDs within the circular Bragg grating cavity for the actual measurements? The connection between the schematics of Fig. 5a and the present samples is not clear.
Moreover, from a general point of view, the authors should pay more attention to guide the reader through the non-obvious reasoning and strategies adopted without forcing to refer at all to the literature. For example, it would be helpful to spend a few words on the advantage of using the first sacrificial QDs for calibration or to comment on the nature of the peak observed in the PL. Last, a carefull proof-reading is still required to improve the language clarity.
Author Response
Please see the attanchment
